# Association between Exposure to Particulate Matter Air Pollution during Early Childhood and Risk of Attention-Deficit/Hyperactivity Disorder in Taiwan

**DOI:** 10.3390/ijerph192316138

**Published:** 2022-12-02

**Authors:** Hueng-Chuen Fan, Chuan-Mu Chen, Jeng-Dau Tsai, Kuo-Liang Chiang, Stella Chin-Shaw Tsai, Ching-Ying Huang, Cheng-Li Lin, Chung Y. Hsu, Kuang-Hsi Chang

**Affiliations:** 1Department of Pediatrics, Tungs’ Taichung Metroharbor Hospital, Wuchi, Taichung 435, Taiwan; 2Department of Rehabilitation, Jen-Teh Junior College of Medicine, Nursing and Management, Miaoli 356, Taiwan; 3Department of Life Sciences, Agricultural Biotechnology Center, National Chung Hsing University, Taichung 402, Taiwan; 4The iEGG and Animal Biotechnology Center, and Rong Hsing Research Center for Translational Medicine, National Chung Hsing University, Taichung 402, Taiwan; 5Ph.D. Program in Translational Medicine, Department of Life Sciences, National Chung Hsing University, Taichung 402, Taiwan; 6Rong Hsing Research Center for Translational Medicine, College of Life Sciences, National Chung Hsing University, Taichung 402, Taiwan; 7School of Medicine, Chung Shan Medical University, Taichung 402, Taiwan; 8Department of Pediatrics, Chung Shan Medical University Hospital, Taichung 402, Taiwan; 9Department of Pediatric Neurology, Kuang-Tien General Hospital, Taichung 433, Taiwan; 10Department of Nutrition, Hungkuang University, Taichung 433, Taiwan; 11Department of Otolaryngology, Tungs’ Taichung MetroHarbor Hospital, Taichung 435, Taiwan; 12Department of Food Science and Biotechnology, National Chung Hsing University, Taichung 402, Taiwan; 13Management Office for Health Data, China Medical University Hospital, Taichung 404, Taiwan; 14Graduate Institute of Clinical Medical Science, China Medical University, Taichung 404, Taiwan; 15Department of Medical Research, Tungs’ Taichung MetroHarbor Hospital, Taichung 435, Taiwan; 16Center for General Education, China Medical University, Taichung 404, Taiwan; 17General Education Center, Jen-Teh Junior College of Medicine, Nursing and Management, Miaoli 356, Taiwan

**Keywords:** attention-deficit/hyperactivity disorder (ADHD), particulate matter (PM), National Health Insurance Research Database (NHIRD)

## Abstract

(1) Background: Recently, a growing number of studies have provided evidence to suggest a strong correlation between air pollution exposure and attention-deficit/hyperactivity disorder (ADHD). In this study, we assessed the relationship between early-life exposure to particulate matter (PM)_10_, PM_2_._5_, and ADHD; (2) Methods: The National Health Insurance Research Database (NHIRD) contains the medical records, drug information, inspection data, etc., of the people of Taiwan, and, thus, could serve as an important research resource. Air pollution data were based on daily data from the Environmental Protection Administration Executive Yuan, R.O.C. (Taiwan). These included particulate matter (PM_2.5_ and PM_10_). The two databases were merged according to the living area of the insured and the location of the air quality monitoring station; (3) Results: The highest levels of air pollutants, including PM_2.5_ (adjusted hazard ratio (aHR) = 1.79; 95% confidence interval (CI) = 1.58–2.02) and PM_10_ (aHR = 1.53; 95% CI = 1.37–1.70), had a significantly higher risk of ADHD; (4) Conclusions: As such, measures for air quality control that meet the WHO air quality guidelines should be strictly and uniformly implemented by Taiwanese government authorities.

## 1. Introduction

Attention-deficit/hyperactivity disorder (ADHD) is a common neurodevelopmental spectrum disorder that has shown a notable increase in incidence over most recent decades. Current epidemiological figures show that ADHD is the most common behavioral disorder in children, affecting approximately 2–7% of children globally [1]. However, data suggest substantial geographical and ethnic variations in the burden of ADHD, with a higher ADHD prevalence in North America than in Europe, but showing no statistical significance between different Diagnostic and Statistical Manual of Mental Disorders editions [2]. In East Asian countries, such as China, Hong Kong, and Taiwan, the prevalence of ADHD is 6.5%, 6.4%, and 4.2%, respectively [3]. ADHD generally develops in preschool and younger children, and tends to show a male predominance. This disorder is characterized by impaired attention, impulsivity, and overactivity, leading to risk-taking behavior, learning difficulties, disorganization, and difficulty completing tasks [4]. Moreover, when the condition persists into adulthood, it can result in a wide range of mental and social difficulties, such as impaired decision-making, antisocial behaviors, emotional dysregulation, employment difficulties, and psychiatric disorders [5,6,7]. Thus, search is ongoing to identify the risk factors and determinants of ADHD to minimize its long-term negative impacts on the development, daily activities, and quality of life of patients and their families.

Although the exact pathogenic mechanisms leading to ADHD are still poorly understood, several genetic and environmental risk factors have been implicated in the development of ADHD-related behavioral abnormalities [8,9]. Genome-wide association studies (GWAS) have shown that genetic susceptibility and single-nucleotide polymorphisms (SNPs) significantly increase the risk of ADHD and symptom distribution [10,11,12,13]. Furthermore, in well-controlled trials, environmental factors linked to the risk of ADHD have been identified, including pollutants, low birth weight, advanced maternal age, socioeconomic factors, and psychological deprivation [14]. Recently, a growing number of studies have investigated the effect of air pollution on the risk of ADHD and have provided evidence of a strong correlation between higher air pollution exposure and ADHD development. In large population-based studies, exposure to particulate matter (PM) and ambient gases in early life significantly increased the risk of ADHD, suggesting a significant role of these parameters in triggering the dysregulation of neurotransmitters and functional abnormalities in the front striatal region [15,16,17].

PM, defined as “a complex mixture of extremely small solid particles and liquid droplets that enter air. Once inhaled, these particles can affect the heart and lungs and cause serious health effects”, is a major component of indoor and outdoor air pollution [18]. Along with aerosols and solid particles, PM usually contains toxic chemicals that can exert hazardous health effects [19]. These particles include PM_2.5_ and PM_10_, classified according to their aerodynamic diameter. PM_10_ is defined as PM with a diameter of less than or equal to 10 μm; this pollution can be absorbed by alveoli into the bloodstream and distributed to several organs. PM_2.5_ is defined as PM with a diameter of less than or equal to 2.5 μm; this pollution can induce cardiopulmonary impairments and/or disorders, and, thus, poses a significant risk to health. A cumulative body of evidence has shown that PM (PM_2.5_ or PM_10_) significantly contributes to the development of a wide range of human diseases, including cardiovascular disorders, chronic obstructive pulmonary disease, pollen allergy, cognitive impairment, malignancy, and all-cause mortality [19,20,21].

To date, several reports have investigated the association between PM exposure (PM_10_ or PM_2.5_) and ADHD [22,23,24]; however, only the impact of PM exposure (PM_10_ and PM_2.5_) in early life has been reported in the literature. In addition, most studies were conducted in Europe, which has a relatively lower PM_2.5_ concentration than Asia, and these studies did not investigate the dose–response relationship between PM_10_ or PM_2.5_ and ADHD. Thus, in this population-based study, we assessed the relationship between early-life exposure to PM_10_ or PM_2_._5_ and ADHD.

## 2. Materials and Methods

### 2.1. Data Source

The National Health Insurance Research Database (NHIRD) contains the medical records, drug information, inspection data, etc., of the people of Taiwan, and, thus, can serve as an important research resource. Disease coding in this database follows the International Classification of Diseases, Ninth Revision, Tenth Revision, and Clinical Modification (ICD-9-CM and ICD-10-CM). The present study used data from the NHIRD, which includes a population of two million. The Taiwan Air Quality-Monitoring Database (TAQMD) contains daily concentrations of PM_2.5_ and PM_10_ from 1998 to 2018, and is maintained by the Environmental Protection Administration Executive Yuan, R.O.C. (Taiwan). The TAQMD includes 74 air quality monitoring stations, which were established based on population density in Taiwan. The two databases were merged according to the living area of the insured and the location of the air quality monitoring station. Each participant was assigned pollutant exposure concentrations based on the data collected from the monitoring station located in their living area.

This study was approved by the institutional review board of the Research Ethics Committee of the China Medical University Hospital (CMUH109-REC2-031(CR-2)).

### 2.2. Study Population, Outcome, and Comorbidities

We investigated the population under the age of 18 in 2003 and living in areas with air quality monitoring stations. The study period began in 2003 and ended in 2017. Attention-deficit hyperactivity disorder (ADHD) (ICD-9-CM:314; ICD-10-CM: F90) was defined in participants having two or more outpatient diagnoses or one admission record. We calculated the annual average amount of air pollutants each subject was exposed to during the follow-up period. The quartiles of each air pollutant were: PM_2.5_ (Q1: <25.5 μg/m^3^, Q2: 25.5–26.4 μg/m^3^, Q3: 26.5–34.2 μg/m^3^, Q4: >34.2 μg/m^3^) and PM_10_ (Q1: < 46.0 μg/m^3^, Q2: 46.0–50.7 μg/m^3^, Q3: 50.8–60.4 μg/m^3^, Q4: >60.4 μg/m^3^). Related comorbidities, such as asthma, atopic dermatitis, and allergic rhinitis, were considered covariates.

### 2.3. Statistical Analysis

The variables sex, age, urbanization level, comorbidities, and ADHD were presented as counts and percentages. Variables related to environmental factors, including temperature and air pollutants PM_2.5_ and PM_10_, were expressed as the mean and standard deviations. The incidence of ADHD per 1000 person-years was calculated. To reveal the association between air pollutants and ADHD in children aged younger than 18 years, the Cox proportional hazards model was used to estimate the risk of ADHD. The incidence rate of ADHD was estimated according to different pollutant levels.

A multivariable model adjusted for covariates was used to estimate the adjusted hazard ratio (aHR) and 95% confidence interval (CI) of the association between the risk of ADHD and PM exposure. Cumulative incidence curves of ADHD were further drawn using the Kaplan–Meier method. Curves were evaluated using the log-rank test. All statistical analyses were performed using SAS software (version 9.4; SAS Institute Inc., Cary, NC, USA). The statistical significance was set at 0.05.

## 3. Results

As shown in Table 1, our study sample consisted of 98,177 participants, 49.3% of whom were female and 50.7% were male. The mean age was 9.66 (±4.27) years, and the highest urbanization level (level one) accounted for 61.4% of the total. Furthermore, 21.4% of patients had the comorbidities of asthma, 9.91% had atopic dermatitis (AD), and 56.2% had allergic rhinitis (AR). The mean temperature was approximately 23.6 (±1.35). The mean annual PM_2.5_ and PM_10_ exposures during the participation period were 29.3 (±7.15) μg/m^3^ and 53.6 (±12.1) μg/m^3^, respectively. Ultimately, 2856 participants developed ADHD. The mean follow-up time was 14.7 (±1.86) years.

Table 2 presents the incidence rates and hazard ratios (95% CI) of ADHD for PM_2.5_ and PM10 stratified into quartiles. The incidence rates of ADHD for the four quartiles of the PM_2.5_ level were 1.15 (Q1), 0.22 (Q2), 2.25 (Q3), and 2.03 (Q4). The crude HR (95% CI) of ADHD in Q2 of PM_2.5_ exposure was 0.19 (0.15, 0.25). Patients in Q3 (cHR = 1.96; 95% CI = 1.75, 2.19) and Q4 (cHR = 1.77; 95% CI = 1.56, 1.99) had a higher risk of ADHD. The incidence rates of ADHD in the four quartiles of the PM_10_ level were 1.45 (Q1), 1.39 (Q2), 2.87 (Q3), and 2.30 (Q4). For PM_10_ pollutants, the subjects in the Q3 level had a 1.96-fold risk of ADHD (95% CI = 1.77, 2.18) and the subjects in the Q4 level had a 1.58-fold risk of ADHD (95% CI = 1.42, 1.76).

Table 3 shows the results of two multivariable models of ADHD, which were adjusted for age, gender, asthma, and AD, with one additionally adjusted for PM_2.5_ and the other one for PM_10_. The greater the age of the participants, the lower the risk of ADHD. In the first model with PM_2.5_, the risk factors of ADHD included male gender (aHR = 2.98; 95% CI = 2.70, 3.29), asthma (aHR = 1.25; 95% CI = 1.13, 1.37), AD (aHR = 1.13; 95% CI = 1.00, 1.28), the Q3 group of PM_2.5_ (aHR = 1.90; 95% CI = 1.70, 2.13), and the Q4 group of PM2.5 (aHR = 1.79; 95% CI = 1.13, 1.37). Children in the Q2 level of PM_2.5_ (aHR = 0.20; 95% CI = 0.15, 0.26) had a lower risk of ADHD than those in the Q1 level. For the other model with PM_10_, male gender (aHR = 3.22; 95% CI = 2.95, 3.51), asthma (aHR = 1.21; 95% CI = 1.11, 1.32), and exposure to high levels of PM_10_ (Q3: aHR = 2.02; 95% CI = 1.82, 2.25; Q4: aHR = 1.53; 95% CI = 1.37, 1.70) all increased the risk of ADHD. The aHR of ADHD for patients with allergic rhinitis (AR) relative to AR-free patients was 0.84 (95% CI = 0.78, 0.91).

As shown in Figure 1, the Q3 and Q4 levels of PM_2.5_ had a relatively high cumulative incidence of ADHD, and the Q2 level of PM_2.5_ had a relatively low cumulative incidence of ADHD when considering the Q1 level of PM_2.5_ as a reference (the *p*-value of the log-rank test was < 0.001). A similar cumulative incidence of ADHD for pollutants PM_10_ was shown.

## 4. Discussion

The association between airborne PM (PM_10_ and PM_2.5_) exposure and human diseases is well-established in the literature [25]. Several regulatory guidelines exist to minimize indoor and outdoor air pollution, including the World Health Organization (WHO) and European Union (EU) air quality guidelines [26]. Unfortunately, an increasing number of studies have demonstrated a significant association between PM exposure and a wide range of disorders, highlighting the suboptimal implementation of these guidelines and the need to reconsider new thresholds for the maximum daily concentration of PM exposure [24]. ADHD is a neurobiological spectrum disorder with multiple underlying pathophysiological mechanisms and several genetic, environmental, and anatomical factors [27]. Researchers have suggested that prenatal and early life exposure to PM significantly increases the risk of ADHD. However, most studies were conducted in Europe and did not simultaneously assess exposure to PM_10_ and PM_2.5_. Thus, in this population-based study, we assessed the relationship between early-life exposure to PM_10_ or PM_2.5_ and ADHD in a Taiwanese population.

Our analysis confirmed the association between higher daily exposure to PM_10_ or PM_2.5_ in early life and the risk of ADHD. We found that, compared with the lower concentrations, the higher concentrations of PM_10_ and PM_2.5_ were significantly associated with a higher incidence of ADHD. These associations remained significant after controlling for age, sex, urbanization level, and comorbidities. In line with our findings, a systematic review of 28 studies by Aghaei et al. reported a significant association between ADHD and higher exposure to air pollutants, including PM_10_ and PM_2.5_ [15]. Another study also found positive associations between prenatal and postnatal exposure to PM_2.5_ and ADHD [22]. In a cohort study in South Korea, every 1 μg/m^3^ increase in exposure to PM_10_ resulted in a two-fold increased risk of ADHD [23]. Exposure to PM_10_ was reported to possibly be a risk factor for worsening ADHD symptoms, resulting in hospitalization [28]. These findings are in line with reports from Mexico, Spain, and India, showing a significant association between exposure to PM and cognitive impairment, low school performance, reduced attention, and behavioral problems [29,30,31]. In this study, the relationship between PM exposure and the risk of ADHD may not have been a dose–response effect; thus, there may be a potential threshold. Further animal or experimental studies are required to confirm these results.

The exact mechanistic approaches underpinning the association between PM exposure and ADHD remain unclear. PM_2.5_ can induce oxidative stress [32], transduce proinflammatory signals from the cardiopulmonary and hepatic systems to the central nervous system [33], and penetrate the BBB to initiate neuroinflammation [34]. PM_2.5_ generated from diesel exhausts is even more toxic, because these air pollutants can absorb polycyclic aromatic hydrocarbons (PAHs) and nitro-substituted PAHs (nitro-PAHs), which both enhance inflammatory responses [35]. PM_10_ can trigger the release of proinflammatory cytokines [36], while PM_10_ and PM_2.5_ can both damage dopaminergic cells [37,38]. Therefore, it is hypothesized that inhaled PM_10_ and PM_2.5_, which can be distributed to various organs, including the brain, via the nasal olfactory mucosa [39], trigeminal nerve route [40], and circulation [33], directly or indirectly induce neurotoxicity, neuroinflammation, and oxidative stress, eventually leading to an increased risk of neurobehavioral disorders.

However, it is worth noting that the current literature shows conflicting results regarding the association between PM exposure and the risk of ADHD. In a recent systematic review by Zhang et al., PM exposure was not significantly associated with a significant increase in the risk of ADHD; the pooled analysis showed substantial heterogeneity [41]. Likewise, a birth cohort study of seven European countries showed no significant association between PM_2.5_ exposure during pregnancy and ADHD [42]. A cohort study from Sweden further reported similar results regarding exposure to PM_10_ [43]. In our opinion, several factors could explain the discrepancies between our results and those obtained in the European population. First, Europe has relatively lower PM_2.5_ and PM_10_ concentrations than Asia; thus, the difference in the exposure to PM between ADHD and non-ADHD individuals may be too small to detect statistical differences. Methodologically, published reports have utilized different exposure assessment tools and diagnostic criteria for ADHD, leading to substantial variability in the estimation or results. On the other hand, ADHD is a multifactorial disorder, and several confounders may contribute to disease development. It is challenging to control for all potential confounders using population-based registries that affect the association between PM exposure and ADHD. Furthermore, the components of air pollution are extraordinarily heterogeneous; hence, the association between PM and ADHD may be attributed to the synergistic effect of the mixture of pollutants, such as NO_2_, PAHs, and nitro-PAHs, rather than solely to PM_2.5_ and PM_10_ concentrations.

To our knowledge, only a few studies have investigated the dose-dependent relationship between early-life exposure to PM_2.5_ and PM_10_ with ADHD. This study collected data from a prospectively collected population-based database, which enhanced the generalizability of our findings. We further explored the association between PM exposure and ADHD after controlling for potential confounders, such as age and socioeconomic status. More importantly, we assessed PM exposure using both temporal and spatial resolution to reduce the risk of exposure misclassification. Additionally, patients were followed-up with for an average of 14.7 years.

Despite its several strengths, we acknowledge the limitations of this study. Firstly, data were retrieved from electronic medical records, which increased the risk of misclassification bias, particularly with a spectrum disorder such as ADHD, in which the diagnosis is challenging due to the substantial variations in presenting symptoms and the extent of impairment. Although we collected data on potential confounders, we could not obtain data regarding ADHD phenotypes and genetic risk factors, precluding the assessment of gene–phenotype–environment interactions. In addition, data regarding other pollutants, such as noise pollution and NO_2_, were not captured; thus, we could not exclude possible interactions with other factors that may have influenced the association between ADHD and PM exposure. Moreover, data regarding relocation during the follow-up period were unavailable; hence, we could not account for changes in PM exposure according to residential relocation. Consequently, we enrolled 98,177 children in this nationwide study. The residential address was not available from the NHIRD. However, the nonresidential geographic contexts (where air pollution affects people) could lead to erroneous exposure assessments [44,45,46]. Most of the participants were preschool and primary school children (mean age, 9.66 ± 4.27 years). Thus, the impact of the neighborhood effect averaging might have been minimized.

## 5. Conclusions

The present population-based study provides further evidence that early-life exposure to PM_2.5_ and PM_10_ significantly increases the risk of developing ADHD even after controlling for common confounders, such as age, socioeconomic status, and comorbidities. These results align with previous toxicological and epidemiologic research showing a negative impact of PM on the behavioral and cognitive development of the population. We further demonstrated a novel finding of the dose–response relationship between PM_2.5_ and PM_10_ and ADHD.

Although regulatory actions and policies to mitigate exposure to air pollution have been implemented in Taiwan, the application of these policies is still suboptimal, and the hazardous health impacts of PM are still relevant. Therefore, measures for air quality control that meet the WHO air quality guidelines should be strictly and uniformly implemented by the Taiwanese government authorities. Community measures that aim to reduce indoor pollution may include preventing indoor smoking, limiting the use of open fireplaces, and providing houses with effective electrostatic pleated air filters or high-quality air purifiers. Previous reports have shown that air filters can reduce PM levels by 60% [47]. However, these measures should be coupled with actions and policies implemented by the Taiwanese governmental authorities to mitigate outdoor pollution. Reducing the maximum permissible value of PM is essential to minimize the risk of ADHD and other health hazards.

Further, well-designed and controlled prospective studies are needed to study the association between PM exposure and ADHD using standard ADHD diagnosis and exposure assessment methods.

## Figures and Tables

**Figure 1 ijerph-19-16138-f001:**
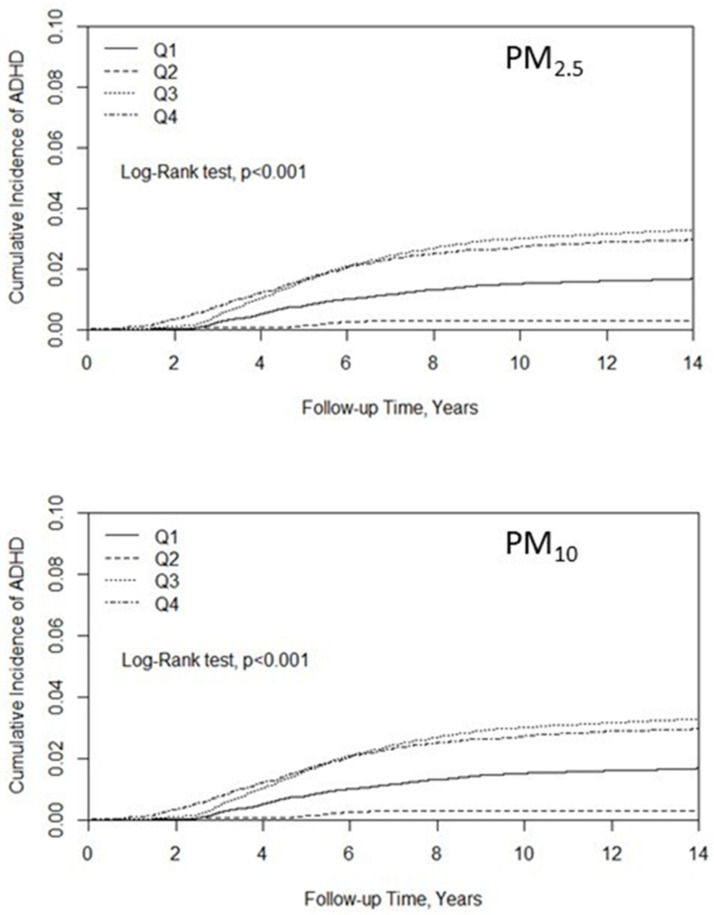
Kaplan–Meier curves of the accumulative incidence rate of ADHD during the follow-up period among the different quartiles for each air pollutant.

**Table 1 ijerph-19-16138-t001:** Baseline demographics and exposure to air pollutants determined using yearly average concentration in Taiwan, 2003–2017.

N = 98,177	Subgroups	*n*	%
Male		49,740	50.7
Age, years	Mean ± SD	9.66 ± 4.27	
Urbanization level	1 (highest)	60,260	61.4
	2	31,108	31.7
	3	6046	6.16
	4 (lowest)	763	0.78
Asthma		21,024	21.4
AD		9728	9.91
AR		55,127	56.2
Temperature	Mean ± SD	23.6 ± 1.35	
PM_2.5_ (yearly average, μg/m^3^)	Mean ± SD	29.3 ± 7.15	
PM_10_ (yearly average, μg/m^3^)	Mean ± SD	53.6 ± 12.1	
ADHD	Yes	2856	2.91
Follow-up time, years	Mean ± SD	14.7 ± 1.86	

Urbanization level was categorized by the population density of the residential area into four levels, with level 1 as the most urbanized and level 4 as the least urbanized. AD, atopic dermatitis; AR, allergic rhinitis; ADHD, attention-deficit hyperactivity disorder; SD, standard deviation.

**Table 2 ijerph-19-16138-t002:** The risk of ADHD in patients exposed to various air pollutants stratified by quartile of daily average concentration using the Cox proportional hazard regression.

Pollutant Levels	N of ADHD	IR	cHR	(95% CI)
PM_2.5_ (μg/m^3^)				
Quartile 1, <25.5	426	1.15		
Quartile 2, 25.5–26.4	62	0.22	0.19	(0.15, 0.25)
Quartile 3, 26.5–34.2	1045	2.25	1.96	(1.75, 2.19)
Quartile 4, >34.2	656	2.03	1.77	(1.56, 1.99)
PM_10_ (μg/m^3^)				
Quartile 1, <46.0	537	1.45		
Quartile 2, 46.0–50.7	513	1.39	0.96	(0.85, 1.08)
Quartile 3, 50.8–60.4	962	2.87	1.96	(1.77, 2.18)
Quartile 4, >60.4	844	2.30	1.58	(1.42, 1.76)

N of ADHD, number of patients with ADHD; IR, incidence rate (per 1000 person-years); cHR, crude hazard ratio; aHR, adjusted hazard ratio; CI, confidence interval. Daily average air pollutant concentrations were categorized into four groups based on the quartiles for each air pollutant.

**Table 3 ijerph-19-16138-t003:** The risk of ADHD in children exposed to PM2.5/PM10 stratified by quartile of daily average concentration using the Cox proportional hazard regression.

Covariates		PM_2.5_		PM_10_	
		aHR	95% CI	aHR	95% CI
Age		0.77	(0.76, 0.78)	0.79	(0.78, 0.80)
Gender	Female	1.00		1.00	
	Male	2.98	(2.70, 3.29)	3.22	(2.95, 3.51)
Urbanization level	1 (highest)	1.00		1.00	
	2	0.90	(0.82, 0.99)	0.97	(0.90, 1.06)
	3	1.01	(0.85, 1.20)	0.94	(0.80, 1.10)
	4 (lowest)	1.22	(0.81, 1.85)	1.23	(0.86, 1.75)
Asthma	No	1.00		1.00	
	Yes	1.25	(1.13, 1.37)	1.21	(1.11, 1.32)
AD	No	1.00		1.00	
	Yes	1.13	(1.00, 1.28)	1.09	(0.98, 1.21)
AR	No	1.00		1.00	
	Yes	0.95	(0.87, 1.04)	0.84	(0.78, 0.91)
Pollutants	Quartile 1	1.00		1.00	
	Quartile 2	0.20	(0.15, 0.26)	0.95	(0.84, 1.07)
	Quartile 3	1.90	(1.70, 2.13)	2.02	(1.82, 2.25)
	Quartile 4	1.79	(1.58, 2.02)	1.53	(1.37, 1.70)

AD, atopic dermatitis; AR, allergic rhinitis; aHR, adjusted hazard ratio, adjusted for age, sex, urbanization level, comorbidities of asthma, AD, and AR; CI, confidence interval.

## Data Availability

Data are available from the NHIRD published by the Taiwan National Health Insurance Bureau. Due to the ‘Personal Information Protection Act’, data cannot be made publicly available (http://nhird.nhri.org.tw/en/index.html; accessed on 1 December 2021).

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
