# Peer review of "Association between Exposure to Particulate Matter Air Pollution during Early Childhood and Risk of Attention-Deficit/Hyperactivity Disorder in Taiwan"

_ijerph, 2022, doi:10.3390/ijerph192316138_

Round 1

Reviewer 1 Report

This is an interesting manuscript that investigates the relationship between air pollution exposure and the risk of ADHD. I have two major comments related to the method part. 

First, the authors need to provide more detailed information on how they proceed with air pollution exposure assessment. The "Data Source" part in its current form does not provide sufficient scientific information on their air pollution exposure assessment methods. Please provide more detailed information on how the authors estimated people's exposure levels. 

Second, in the field of pollution exposure assessment studies, it is widely known that overlooking non-residential geographic contexts (where air pollution affects people) can lead to erroneous exposure assessments. However, it seems that this study does not consider non-residential geographic context when they delineate geographic context where air pollution influences their samples. Please refer to the following papers on this important methodological issue and discuss the potential limitations of the authors' method.

Kim, J., & Kwan, M. P. (2021). How neighborhood effect averaging might affect assessment of individual exposures to air pollution: A study of ozone exposures in Los Angeles. Annals of the American Association of Geographers, 111(1), 121-140.

Kwan, M. P. (2012). The uncertain geographic context problem. Annals of the Association of American Geographers, 102(5), 958-968.

Setton, E., Marshall, J. D., Brauer, M., Lundquist, K. R., Hystad, P., Keller, P., & Cloutier-Fisher, D. (2011). The impact of daily mobility on exposure to traffic-related air pollution and health effect estimates. Journal of exposure science & environmental epidemiology, 21(1), 42-48.

Author Response

First, the authors need to provide more detailed information on how they proceed with air pollution exposure assessment. The "Data Source" part in its current form does not provide sufficient scientific information on their air pollution exposure assessment methods. Please provide more detailed information on how the authors estimated people's exposure levels.

Reply

We thank you for the suggestion and we have re-written the paragraph, as follows:

The Taiwan Air Quality-Monitoring Database (TAQMD), contains daily concentrations of PM2.5 and PM10 from 1998 to 2018, and is maintained by the Environmental Protection Administration Executive Yuan, R.O.C. (Taiwan). TAQMD has included 74 air quality monitoring stations, which were established by population density in Taiwan. The two databases were merged according to the living area of the insured and the location of the air quality monitoring station. Each participant was assigned pollutant-exposure concentrations based on the data collected from the monitoring station located in their living area.

Second, in the field of pollution exposure assessment studies, it is widely known that overlooking non-residential geographic contexts (where air pollution affects people) can lead to erroneous exposure assessments. However, it seems that this study does not consider non-residential geographic context when they delineate geographic context where air pollution influences their samples. Please refer to the following papers on this important methodological issue and discuss the potential limitations of the authors' method.

Reply

We thank you for the suggestion and we have added the paragraph in the section of discussion, as follows:

Consequently, we enrolled 98177 children in this nationwide study. The residential address was not available from the NHIRD. However, the non-residential geographic contexts (where air pollution affects people) can lead to erroneous exposure assessments. Most of the participants were pre-school and primary school children (mean age, 9.66 years). Thus, the impact of the neighborhood effect might be minimized by averaging the data.

Reviewer 2 Report

1. The title can specify Taiwan as the studied location/ population

2. More descriptions and references can be provided in the statistical analysis section to explain why the crude hazard ration and the Cox proportional hazards model were adopted.

3. Table 1 index is unclear. The columns “n” and “%” sometimes mean “number” and “percentage”, sometimes mean “mean” and “SD”. Please amend the table.

4. There is a mismatch of data in line 165 and table 2 regarding the crude HR in Q3 of PM10.

5. Any possible explanation can be provided on the fact that children in the Q2 level of PM2.5 had a lower risk of ADHD than those in the Q1 level?

6. The discussion on dose-response relationship between PM2.5 and PM10 seems to be missing. Dose-response relationship shall describe the magnitude of the response, in this case, may be the risk of developing ADHD, as a function of exposure to PM.

7. The first line in the conclusion should not be there. Please check the rest of the manuscript carefully.

Author Response

Reviewer 2

  1. The title can specify Taiwan as the studied location/ population

Reply

We thank you for the suggestion and we have modified the title, as follows:

Association between exposure to Particulate Matter Air Pollution During Early Childhood and Risk of Attention-deficit/Hyperactivity Disorder in Taiwan.

  1. More descriptions and references can be provided in the statistical analysis section to explain why the crude hazard ration and the Cox proportional hazards model were adopted.

Reply

We thank you for the suggestion and we had re-written the paragraph, as follows:

The incidence of ADHD per 1000 person-years was calculated. To reveal the association between air pollutants and ADHD in children aged younger than 18 years, the Cox proportional hazards model was used to estimate the risk of ADHD. The incidence rate of ADHD was estimated according to different pollutant levels. A multivariable model adjusted for covariates was used to estimate the adjusted hazard ratio (aHR) and 95% confidence interval (CI) of the association between the risk of ADHD and PM exposure.

  1. Table 1 index is unclear. The columns “n” and “%” sometimes mean “number” and “percentage”, sometimes mean “mean” and “SD”. Please amend the table.

Reply

We thank you for the suggestion and we have modified table 1.

  1. There is a mismatch of data in line 165 and table 2 regarding the crude HR in Q3 of PM10.

Reply

We thank you for the suggestion and we have corrected the crude HR in Q3 of PM10.

  1. Any possible explanation can be provided on the fact that children in the Q2 level of PM2.5 had a lower risk of ADHD than those in the Q1 level?

Reply

The lower risk in the Q2 level might be due to bias because of the fewer cases of ADHD.

  1. The discussion on dose-response relationship between PM2.5 and PM10 seems to be missing. Dose-response relationship shall describe the magnitude of the response, in this case, may be the risk of developing ADHD, as a function of exposure to PM.

Reply

We thank you for the suggestion and we had added the paragraph, as follows:

In this study, the relationship between PM exposure and the risk of ADHD may not be a dose-response effect; thus, there may be a potential threshold. Further animal or experimental studies are required to confirm these results.

  1. The first line in the conclusion should not be there. Please check the rest of the manuscript carefully.

Reply

We thank you for the suggestion and we have modified it.

Round 2

Reviewer 1 Report

I appreciate the authors' efforts in addressing reviewers' comments.